# Genetic Diversity and Spatiotemporal Distribution of SARS-CoV-2 Alpha Variant in India

**Jahnavi Parasar** [1,2,†]**, Rudra Kumar Pandey** [1,†]**, Yashvant Patel** [1,†]**, Prajjval Pratap Singh** [1]**, Anshika Srivastava** [1]**, Rahul Kumar Mishra** [1]**, Bhupendra Kumar** [3]**, Niraj Rai** [4]**, Vijaya Nath Mishra** [5]**, Pankaj Shrivastava** [6]**, P. B. Kavi Kishor** [7]**, Prashanth Suravajhala** [8]**, Rakesh Tamang** [9]**, Ajai Kumar Pathak** [10] **and Gyaneshwer Chaubey** [1,*]

1   Cytogenetic Laboratory, Department of Zoology, Banaras Hindu University, Varanasi 221005, India
2   Department of Biotechnology, Institute of Health and Allied Sciences, Ghaziabad 201206, India
3   Entomology Laboratory, Department of Zoology, Banaras Hindu University, Varanasi 221005, India
4   Birbal Sahni Institute of Palaeosciences, Lucknow 226007, India
5   Department of Neurology, Institute of Medical Sciences, Banaras Hindu University, Varanasi 221005, India
6   Biology and Serology Division, Department of Home (Police), Government of Madhya Pradesh, Regional Forensic Science Laboratory, Bhopal 462003, India
7   Department of Genetics, Osmania University, Hyderabad 500007, India
8   Amrita School of Biotechnology, Amrita Vishwa Vidyapeetham, Clappana 690525, India
9   Department of Zoology, University of Calcutta, Kolkata 700073, India
10  Institute of Genomics, University of Tartu, 51010 Tartu, Estonia
*   Correspondence: gyaneshwer.chaubey@bhu.ac.in
†   These authors contributed equally to this work.

**Abstract:** After the spill to humans, in the evolutionary timeline of SARS-CoV-2, several positively selected variants have emerged. A phylogeographic study on these variants can reveal their spatial and temporal distribution. In December 2020, the alpha variant of the severe acute respiratory syndrome coronavirus (SARS-CoV-2), which has been designated as a variant of concern (VOC) by the WHO, was discovered in the south-eastern United Kingdom (UK). Slowly, it expanded across India, with a considerable number of cases, particularly in North India. This study focuses on determining the prevalence and expansion of the Alpha variants in various parts of India mainly by using phylospatial analysis. The genetic diversity estimation has helped us to understand various evolutionary forces that have shaped the spatial distribution of this variant during its peak. Overall, our study paves the way to understanding the evolution and expansion of a virus variant, which may help to mitigate in the case of any future wave.

**Keywords:** SARS-CoV-2; alpha variant; phylogenetics; founder effect

## 1. Introduction

The COVID-19 (Coronavirus Disease 2019) pandemic, caused by the SARS Corona virus 2 (SARS-CoV-2), has impacted the world, with India having the second-highest number of confirmed cases (MOHFW-GoI, n.d.) [1]. The first instances of COVID-19 in India were recorded in Kerala in January 2020 among three medical students who had returned from Wuhan, the pandemic's first epicentre [2,3]. SARS-CoV-2 is a single-stranded, positive-sense RNA (+ssRNA) virus in the Coronaviridae family that belongs to lineage B of the genus beta coronavirus. The genome size of SARS-CoV-2 is ~29.9 kb, sharing ~78% sequence homology with SARS-CoV [4]. SARS-CoV-2 has been evolving at a rate consistent with the virus gaining two mutations per month in the global population since December 2019 [5]. Mutation has given rise to different variants of SARS-CoV-2. Any virus strain becomes a variant of concern when it increases transmissibility and with virulence or decreases its effectiveness towards vaccines [6].

The Alpha VOC (Variant of Concern) (lineage B.1.1.7) first appeared in the United Kingdom [7] the Beta VOC (lineage B.1.351) in South Africa, and the Gamma VOC (lineage B.1.1.28.1 or P.1) in Brazil [8], and the Delta VOC (lineage B.1.617.2) in India [9]. These VOCs were thought to be more resistant to the neutralizing activity of anti-bodies produced during natural infection or vaccination and their increased transmission rate [10]. The viral spike protein's receptor binding domain (RBD) contains most of the mutations observed in VOCs [11]. Mutation N501Y, present in all except the Delta VOC, confers a higher affinity for the cellular ACE2 (angiotensin-converting enzyme 2) viral receptor and may be related to the increased transmissibility of VOCs carrying this mutation [12]. After being detected in the UK in no time, the Alpha variant became dominant in other countries. Its rapid success was due to its ability to weaken our body's first immune defence line, giving the mutant more time to reproduce. Alpha variant has 23 mutations, 8 of which are in the spike protein, that distinguish it from other coronaviruses. N501Y, spike deletion 69–70del, and P681H are the three mutations that have the greatest biological impact. It was also found that there was an 80-fold increase in Orf9b, which leads to an increase in Orf9b protein levels in this variant. The Orf9b protein of the virus antagonises the innate immune response by interacting with Tom 70, a mitochondrial import receptor required for Type I interferon production [13]. There is evidence to suggest that the Beta (B.1.351) and Delta (B.1.617.2) variants of SARS-CoV-2 may be associated with reduced interferon production compared to the Alpha (B.1.1.7) variant [14].

According to the data obtained from GISAID [15], there is a variation in the number of Alpha cases recorded in different parts of India. When the frequency of the Alpha variant was calculated across India, it was found that it is more common in various northern and central Indian states than in the southern states. In February and March 2020, several Indian states had a sudden spike in COVID-19 cases largely driven by the Alpha variant, after which India saw the second wave of the pandemic. The phylogenetic examination of samples from multiple Indian states revealed that it first entered the northern portions of the country, from where it spread throughout the country. Affected regions vary in their genetic diversity estimates. The relationship between genetic diversity and the number of cases has provided an idea of expansion and spread that prevailed in those regions, resulting in variable diversity in different regions.

## 2. Materials and Methods

GISAID, an open source for COVID-19 and influenza viruses, was used to download the genomic data of the Alpha variant [15]. We have downloaded only complete and high-quality sequences. MAFFT [16] was used to align genome sequences to reference sequences. Aligned sequences were downloaded in Fasta format. Using these aligned Fasta sequences, we constructed a maximum likelihood tree using MEGAX [17], using the Taimura Nei Model and Tree Inference options as the nearest neighbour interchange, complete deletion for gap and missing data treatment, and the rest parameters were kept as default.

Charts and graphs were constructed by (Google Sheets, 2009; https://www.google.com/sheets/about/) [18]. Frequency maps were generated using Datawrap-per (2012) (https://www.datawrapper.de/) [19], and the haplotype diversity was derived from DnaSP [20].

A total of 1131 high-quality genomic sequences of the alpha variant were extracted on 11 June 2021 with a high coverage and were aligned to the reference sequence (Wuhan/2019-EPI_ISL_402124). Each state, union territory, and Wuhan sequences were labelled with distinct colours to be distinguished on a phylogenetic tree. The Alpha variant samples were filtered from all the affected regions, and the frequency was calculated. Then, from December 2020 to May 2021, the frequency was determined separately for each month and distribution in different parts of India was evaluated. Its dominance in specific locations of India was determined using the obtained frequency. The samples from each state and union

territory were analysed on DnaSP [21] to determine mutations in all impacted regions. The value of haplotype diversity was calculated using Fu and Li's approach [22].

## 3. Results and Discussion

India has seen three distinct COVID-19 waves [23]. Among these three waves, the second wave was most devastating [24]. Due to the high infectivity of new virus variants (mainly Alpha, Beta, and Delta in India), a large number of people contracted the virus, which overwhelmed the health system, and many people died as a result of having no place to stay in the hospitals. The surge in cases started in mid-March 2021 and rapidly peaked with nearly 400,000 cases per day during the first week of May 2021. The genome sequencing data in the timeline showed that it was mainly driven by Alpha, Beta, and Delta variants. Thus, we have taken the phylogeographic study of the Alpha variant in the present study to understand the phylogeography and expansion of this variant in the country.

Among the available sequences of this variant from India, Punjab state had the highest number with a frequency of 0.740, followed by Chandigarh (0.467) and Maharashtra (0.296) (Supplementary Table S1). According to GISAID data, cases of Alpha variant in India began to appear in a few states during October and November 2020, while the UK's first sample was reported in September 2020 [25]. In November 2021, samples of Alpha were discovered in the states of Delhi, Gujarat, and Jammu and Kashmir, and it gradually spread to other regions of India, thus suggesting multiple independent entries of this variant to India from the UK.

In our phylogenetic analysis, we observed two major divisions for the Alpha variant in India (Figure 1). Geographically, both of these divisions are prevalent in North India. The phylogenetic tree also showed at least three different expansion timelines in the country. All these timelines are starlike expansions, suggesting a rapid dispersal. These dispersal events are associated with the overwhelming cases of the Alpha variant in northern India. Our phylogenetic analysis revealed that Punjab and Delhi samples shared a common ancestry at several nodes. These nodes have evolved in parallel, indicating a link between the two regions and possibly the cradle of the virus spread to other areas (Figure 1 and Supplementary Figure S1). Such observations also indicate a frequent movement of people in these two states. Samples from Sikkim, Chhattisgarh, Maharashtra, Gujarat, Jammu and Kashmir, and Uttarakhand shared a common ancestry with Delhi and Punjab (Figure 1). In our phylogenetic tree, there are only three independent branches which are lacking Punjab, Chandigarh, and New Delhi samples.

For the temporal comparison, the month-by-month frequency was computed (Figure 2 and Supplementary Figure S2). None of India's zones had a high frequency of Alpha variant in December and January, while Punjab saw a 0.354 frequency elevation in February. Along with Punjab, Madhya Pradesh experienced a spike in March 2021. Except for the north-east states, practically every region had the highest frequency in March. In March, Punjab had a dramatic increase in frequency (0.6), followed by Chandigarh (0.294), Madhya Pradesh (0.214), and Jammu–Kashmir (0.16) (Figure 2). North-eastern states such as Meghalaya and Nagaland received the Alpha wave in a mediated fashion later in to April 2021.

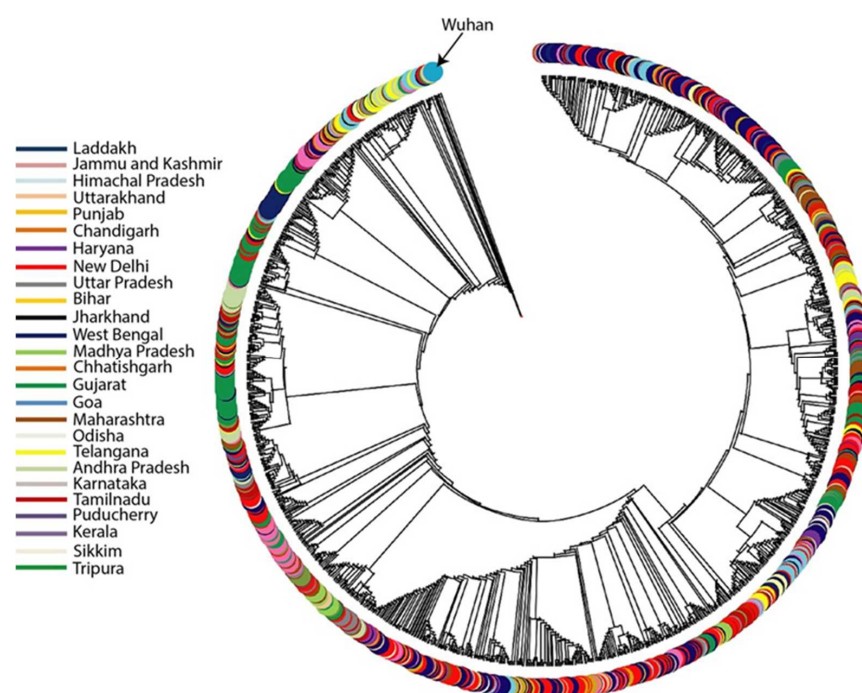

**Figure 1.** The maximum likelihood (ML) tree of SARS-CoV-2 alpha (B1.1.7) variant showing the phylogenetic relationships of all the alpha variant samples reported in India. The samples have been segregated state-wise.

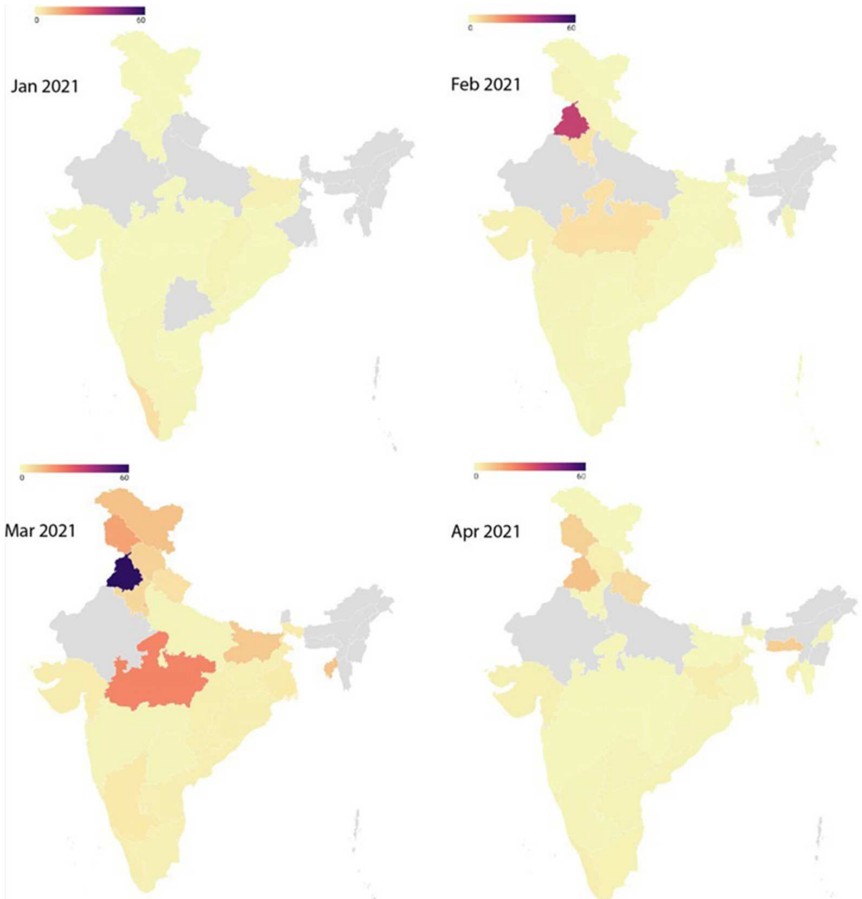

**Figure 2.** The geospatial frequency distribution of Alpha variant in India between January and April 2021.

In order to understand the diversity of this variant in different regions of India, we computed the haplotype diversity (Table 1). The diversity of this variant ranged from 0.798 to 0.997 in India. The highest haplotype diversity values were found in Chandigarh and Haryana (0.997), followed by Gujarat and Delhi, 0.995 (Table 1). This disparity in the range of the diversity values among various locations suggests variable viral dynamics in different regions (Table 1). The high diversity in Chandigarh, Haryana, Gujarat, and New Delhi suggested the early multiple entry and local differentiation of this virus from the UK. After multiple entries, the virus expanded among the population rapidly.

**Table 1.** Haplotype diversity of alpha variant (B1.1.7) in various regions of India. The values are arranged in descending order.

| States and Union Territories | Total Number of Samples | Haplotype Diversity |
|:---:|:---:|:---:|
| Chandigarh | 92 | 0.997 |
| Haryana | 46 | 0.997 |
| Gujarat | 194 | 0.995 |
| New Delhi | 681 | 0.995 |
| Telangana | 74 | 0.993 |
| Andhra Pradesh | 56 | 0.992 |
| Maharashtra | 120 | 0.989 |
| Tamil Nadu | 70 | 0.986 |
| Jammu Kashmir | 189 | 0.985 |
| Uttar Pradesh | 2 | 0.983 |
| Uttarakhand | 66 | 0.982 |
| Odisha | 29 | 0.978 |
| Rajasthan | 3 | 0.976 |
| Puducherry | 35 | 0.974 |
| Madhya Pradesh | 237 | 0.971 |
| West Bengal | 171 | 0.971 |
| Chhattisgarh | 92 | 0.961 |
| Jharkhand | 22 | 0.956 |
| Himachal Pradesh | 48 | 0.948 |
| Assam | 2 | 0.933 |
| Tripura | 5 | 0.915 |
| Karnataka | 131 | 0.895 |
| Sikkim | 8 | 0.863 |
| Punjab | 692 | 0.861 |
| Goa | 5 | 0.8 |
| Meghalaya | 15 | 0.798 |

Irrespective of the highest surge of this variant in Punjab, we do not see any associated high diversity of this variant, similar to Chandigarh and New Delhi (Table 1). This suggests that the virus spread in Punjab was rapid and associated with a limited number of founder events. Additionally, the samples from Punjab are largely restricted in a few clusters rather than independent lineages. Founder events were usually favoured over super-spreader events, e.g., indoor meetings, large gathering, etc. [26].

To understand this rapid surge in Punjab with limited founder events, we have looked at the social gathering which occurred at the given time (Figure 2). The Anti-Farm Law protest [27] was ongoing during that period, with a large involvement of people from Punjab [28]. The pattern of a spike associated with limited founders suggest a likely association of COVID-19 surge with the Anti-Farm Law protest. Some news sources have also investigated this issue and reported that a sudden surge in villages of Punjab was due to returning farmers from the protest site (Anti-Farm Law Protest Report, n.d.).

This indicates that farmers' protests in Punjab and other regions caused an exponential frequency rise in Punjab, with the majority of the farmers hailing from Punjab and Haryana. Rallies were staged in several parts of North India, highlighting the critical role of social dynamics in the SARS-CoV-2 pandemic and the emergence of the second wave in India.

In April, cases of the Delta variant were more prevalent than those of the Alpha type, and the frequency percentage in most locations declined. Concerning the Alpha variant, there existed a strong link between Delhi and Punjab. The outbreak in Delhi in April 2021 was preceded by outbreaks in Maharashtra, Kerala, and Punjab, and we can say that the outbreak in Delhi in 2021 was fuelled by the Alpha variant (GISAID—Initiative, 2008). However, in later months, Alpha was completely wiped out by the more competent variant Delta.

Thus, this study outlines the need for phylogenetics to understand a virus variant's spatial and temporal distribution and role of social affairs. The virus variant Alpha was predominant in North India compared to the southern part of the country. The rapid expansion of this variant with limited founders in Punjab was likely to be associated with the farmers' agitation. Its moderate presence in the south was likely due to a more virulent strain Delta which eventually took hold of the complete landscape of India after that.

## 4. Conclusions

In the course of coronavirus spread and infection, many policies have been administered worldwide to stop the spread. Informal social gatherings have been considered a serious concern, however, there is limited data on the scientific evaluations. In this study, we provide direct evidence on the question of how large social gatherings spread COVID-19. While studying the temporal and spatial spread of the Alpha variant in India, we observed, in some states, a correlation of frequency surge driven by limited founders with the large gathering, rapid movement and mixing of the populations. Our findings are informative to understand the variant spread and more risk of infection to the people associated with large gatherings.

We caution that the conclusion of this study is dependent on the number of sequences and available samples. However, there are places where the proper testing, collecting, sequencing, and submission of samples have not been adequately made, which may impact the outcome.

Overall, it is important to carefully consider the sampling strategy when obtaining viral sequences for variant surveillance in order to ensure that the resulting data accurately reflect the prevalence and distribution of different variants in the population of interest.

**Supplementary Materials:** The following are available online at https://www.mdpi.com/article/10.3390/covid3040035/s1.

**Author Contributions:** G.C. conceived and designed this study. J.P., R.K.P., Y.P., P.P.S., A.S., R.K.M., B.K., N.R., V.N.M., P.S. (Pankaj Shrivastava), P.B.K.K., P.S. (Prashanth Suravajhala), R.T. and A.K.P. collected the data. J.P., R.K.P., Y.P., P.P.S., A.S., P.S., R.T. and A.K.P. analysed the data. J.P., A.S., P.P.S., P.S. and G.C. wrote the manuscript with the input of other co-authors. All authors contributed to the article and approved the submitted version. All authors have read and agreed to the published version of the manuscript.

**Funding:** This research was funded by Indian Council of Medical Research (ICMR) ad hoc grant numbers 2021-6389 and 2021-11289.

**Institutional Review Board Statement:** The study was conducted in accordance with the Declaration of Helsinki, and approved by the Ethics Committee of Banaras Hindu University, Varanasi, India (Ref No. I.Sc./ECM-XII/2021-2022/), for studies involving humans.

**Informed Consent Statement:** Informed consent was obtained from all subjects involved in the study.

**Data Availability Statement:** All datasets generated for this study are included in the article/ Supplementary Materials.

**Acknowledgments:** G.C. and V.N.M. is supported by Faculty IOE grant BHU (6031). P.P.S. is supported by CSIR SRF fellowship. R.K.P. is supported by ICMR- SRF fellowship. A.S. is supported by UGC-CAS fellowship. R.K.M. is supported by Mahamana post-doctoral fellowship BHU and A.K.P. is Supported by the Estonian Research Council fund number PUTJD1186. G.C. and R.T. acknowledge the supported from SERB, Government of India (CRG/2018/001727).

**Conflicts of Interest:** The authors declare no competing interests. The authors declare that the research was conducted in the absence of any commercial or financial relationships that could be construed as a potential conflict of interest.

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
