# Peer review of "Genetic Diversity and Spatiotemporal Distribution of SARS-CoV-2 Alpha Variant in India"

_covid, doi:10.3390/covid3040035_

Round 1

Reviewer 1 Report

After more than two years since the first case associated with SARS-CoV-2 was reported, the appearance of mutations over time has given rise to the emergence of variants of the virus. Alpha, a variant of concern (VOC) according to the WHO, slowly spread across India with a considerable number of cases. The study focuses on determining the prevalence and spread of Alpha variants in various parts of India, mainly through phylospatial analysis. For this, the study shaped the spatial distribution of this variant during one of India's peaks. It also highlights the role of alpha genetic diversity in estimating viral dynamics.

In general, the conclusions are supported by the study methodology. However, I suggest expanding the information a little more regarding the samples included in the study. Specifically, my comments regarding the article are:

·       In line 54, “The viral Spike protein's Receptor 53 Binding Domain (RBD) contains most of the mutations observed in VOCs (S)”. It is not clear at the end of the sentence whether what is in parentheses is a reference or not.

·       In line 85, could you include the parameters to run MEGAX?

·       In line 89, what is the time window in which the sequences were downloaded? Is it since the arrival of alpha to the country?

·       Why didn't you include Beta and Delta in the study if they also had a significant impact on the second wave?

·       In line 118, “we see two major divisions for Alpha variant in India…” I suggest including in the graph the delimitations of the two divisions mentioned by the authors

·       I suggest including a graph of the number of COVID-19 cases by variant in the same time window in which the geospatial frequency of alpha was evaluated, which allows the reader to observe the contribution (in number of cases) of each variant in the mentioned period.

·       In line 182, “Its moderate presence in the South was likely due to a more virulent strain delta which has eventually taken the complete landscape of India after that”. To conclude that the lower prevalence of alpha in the south is due to the early entry of delta, an analysis of the distribution of delta over the same period is necessary to confirm that it had a higher prevalence in the south and that this could have displaced alpha.

·       In line 112, Madhya Pradesh is mentioned as the third state with the highest frequency (0.296) of alpha. However, in Supplementary Table 1, this state is reported to have a frequency of 0.0201

·       In supplementary figure 2 the Y axis is not indicated. Is it the frequency of the variant?

Reviewer 2 Report

This review article has determined the prevalence and expansion of the Alpha variants in various parts of India mainly by sequence analysis.

 Several suggestions:

1.      Line 36, please add the full name for COVID-19 when it is mentioned the first time.

2.      Line 38, 47, 69, 82, 85, 86 etc., please add the website addresses for these references, such as WHO, Google sheet.

3.      Lines 40, 43, etc. please use the same format for the citing references in the entire manuscript.

4.      Line 61, it is better to use [Spike] than [spike] protein.

5.      Line 68, please add a reference after [they do not overproduce Orfb9 proteins].

6.      Line 95, [Its dominance in specific locations of India was discovered using the obtained frequency.]. If the viral sequences are not from all patients, will this approach generate bias?

Round 2

Reviewer 2 Report

Issues raised previously have been addressed in this revised manuscript.